# Hyperprogressive Disease during Anti-PD-1 (PDCD1) / PD-L1 (CD274) Therapy: A Systematic Review and Meta-Analysis

**DOI:** 10.3390/cancers11111699

**Published:** 2019-11-01

**Authors:** Jong Yeob Kim, Keum Hwa Lee, Jeonghyun Kang, Edith Borcoman, Esma Saada-Bouzid, Andreas Kronbichler, Sung Hwi Hong, Leandro Fórnias Machado de Rezende, Shuji Ogino, Nana Keum, Mingyang Song, Claudio Luchini, Hans J. van der Vliet, Jae Il Shin, Gabriele Gamerith

**Affiliations:** 1Yonsei University College of Medicine, Seoul 03722, Korea; crossing96@yonsei.ac.kr; 2Department of Pediatrics, Yonsei University College of Medicine, Seoul 03722, Korea; AZSAGM@yuhs.ac; 3Department of Surgery, Gangnam Severance Hospital, Yonsei University College of Medicine, Seoul 06273, Korea; ravic@naver.com; 4Department of Drug Development and Innovation (D3i), Institut Curie, 75 005 Paris, France; edith.borcoman@curie.fr; 5Department of Medical Oncology, Centre Antoine Lacassagne, University Cote d’Azur, 06189 Nice, France; esma.saada-bouzid@nice.unicancer.fr; 6Department of Internal Medicine IV, Medical University Innsbruck, 6020 Innsbruck, Austria; andreas.kronbichler@i-med.ac.at; 7Department of Global Health and Population, Harvard T. H. Chan School of Public Health, Boston, MA 02115, USA; sunghwihong@gmail.com; 8Universidade Federal de São Paulo, Escola Paulista de Medicina, Departamento de Medicina Preventiva, São Paulo 04023-062, Brazil; leandro.rezende@unifesp.br; 9Broad Institute of MIT and Harvard, Cambridge, MA 02142, USA; SOGINO@BWH.HARVARD.EDU; 10Cancer Immunology and Cancer Epidemiology Programs, Dana-Farber Harvard Cancer Center, Boston, MA 02115, USA; 11Department of Epidemiology, Harvard T.H. Chan School of Public Health, Boston, MA 02115, USA; mis911@mail.harvard.edu; 12Program in MPE Molecular Pathological Epidemiology, Department of Pathology, Brigham and Women’s Hospital, and Harvard Medical School, Boston, MA 02115, USA; 13Department of Food Science and Biotechnology, Dongguk University, Goyang 04620, Korea; nak212@mail.harvard.edu; 14Department of Nutrition, Harvard T.H. Chan School of Public Health, Boston, MA 02115, USA; 15Clinical and Translational Epidemiology Unit, Massachusetts General Hospital and Harvard Medical School, Boston, MA 02115, USA; 16Division of Gastroenterology, Massachusetts General Hospital and Harvard Medical School, Boston, MA 02115, USA; 17Department of Diagnostics and Public Health, Section of Pathology, University of Verona, 37134 Verona, Italy; claudio.luchini@univr.it; 18Department of Medical Oncology, Amsterdam UMC, Cancer Center Amsterdam, VU University, 1081 HV Amsterdam, The Netherlands; JJ.vanderVliet@vumc.nl; 19Internal Medicine V, Department of Hematology & Oncology, Medical University Innsbruck, 6020 Innsbruck, Austria; gabriele.gamerith@i-med.ac.at; 20Tyrolean Cancer Research Institute, 6020 Innsbruck, Austria

**Keywords:** immunotherapy, hyperprogression, hyper-progressive disease

## Abstract

Hyperprogressive disease (HPD) is a recently acknowledged pattern of rapid tumor progression after the initiation of immune checkpoint inhibitors. HPD has been observed across various types of tumors and has been associated with poor survival. We performed a meta-analysis to identify baseline (i.e., prior to programmed cell death 1 [PD-1, PDCD1] / programmed cell death 1 ligand 1 [PD-L1, CD274] inhibitor therapy) patient factors associated with risks of developing HPD during PD-1/PD-L1 inhibitor therapy. We searched eight databases until 6 June 2019. We calculated the summary odds ratio (OR) and its 95% confidence interval (CI) using the random-effects model and explored between-study heterogeneity and small-study effects. A total of nine articles was eligible (217 HPD cases, 1519 cancer patients) for meta-analysis. There was no standard definition of HPD, and the incidence of HPD ranged from 1 to 30%. We identified twenty-three baseline patient factors, of which five factors were statistically significantly associated with HPD. These were serum lactate dehydrogenase (LDH) above the upper normal limit (OR = 1.89, 95% CI = 1.02–3.49, *p* = 0.043), more than two metastatic sites (OR = 1.86, 1.34–2.57, *p* < 0.001), liver metastases (OR = 3.33, 2.07–5.34, *p* < 0.001), Royal Marsden Hospital prognostic score of 2 or above (OR = 3.33, 1.96–5.66, *p* < 0.001), and positive PD-L1 expression status that was inversely correlated with HPD (OR = 0.60, 0.36–0.99, *p* = 0.044). Between-study heterogeneity was low. Evidence of small-study effect was found in one association (PD-L1 expression). Subset analyses of patients with non-small cell lung cancer showed similar results. Future studies are warranted to identify underlying molecular mechanisms and to test their roles as predictive biomarkers of HPD.

## 1. Introduction

Immune checkpoint inhibitors, having been recently included in clinical practice for the treatment of different types of solid tumors such as melanoma [1], non-small cell lung cancer (NSCLC) [2,3], head and neck squamous cell carcinoma (HNSCC) [4], and renal cell carcinoma [5], have demonstrated therapeutic responses across multiple types of cancers. With the immune checkpoint inhibitors, unique patterns of tumor responses to treatment such as delayed response and pseudoprogression were described. Pseudoprogression, characterized by an initial increase in tumor burden or the appearance of new lesions followed by tumor shrinkage, is typically associated with better survival compared to non-responders despite seemingly detrimental findings with initial tumor imaging [6]. The observation of such phenomena has led to the development of tumor response assessment criteria specific for immunotherapy such as immune-related response criteria (irRC), immune-related response evaluation criteria in solid tumors (irRECIST), and modified RECIST 1.1 criteria for immune-based therapeutics (iRECIST) [6,7,8], which require different interpretation and additional imaging to confirm disease progression.

Besides pseudoprogression, another pattern of progression has been described in patients treated with immune checkpoint inhibitors, particularly during treatment with antibodies against programmed cell death 1 (PDCD1, PD-1) or programmed cell death 1 ligand 1 (CD274, PD-L1), called hyperprogression. Hyperprogression, initially described and reported extensively by Champiat and colleagues in 2017 [9], is characterized by a rapid progression of tumor after the initiation of immune checkpoint inhibitor therapies. The existence of hyperprogression has been considered controversial and remains not well defined, but the acceleration of tumor growth exceeding twofold based on three imaging timepoints (pre-treatment, baseline, post/under-treatment) has been most widely used to define hyperprogressive disease (HPD) [9,10].

The phenomenon has been observed across various types of tumors [9,11], and subsequent studies have reported that patients with HPD, in contrast to pseudoprogression, have substantially lower survival rates compared to patients not showing HPD. These findings suggest that anti-PD-1/PD-L1 therapy can play a deleterious role in a subset of patients with various tumors. Ferrara [12], Aoki, and colleagues [13] observed that in patients with NSCLC or advanced gastric cancer (AGC), HPD was more common among anti-PD-1/PD-L1 monotherapy cohorts compared to the chemotherapy cohorts, and that patients with HPD showed poorer prognosis than patients showing progression without HPD in the anti-PD-1/PD-L1 therapy cohorts. These results suggest that HPD during anti-PD-1/PD-L1 therapy should be distinguished from simple disease progression stemming from a lack of response to conventional cytotoxic chemotherapy, and perhaps being directly related to the mechanisms of immune checkpoint inhibition.

Hyperprogression is a new type of tumor response that should be recognized in making treatment decisions for patients with cancer. Clinicians should weigh the potential harm arising from the risk of HPD upon the initiation of anti-PD-1/PD-L1 therapy. Comprehensive quantification of incidence and prognosis and identification of predictive factors of HPD is therefore crucial. Multiple studies of immunotherapy cohorts have reported characteristics of HPD. However, sample sizes were generally small, and results were conflicting across some studies. To synthesize the available evidence and summarize the findings for characteristics of HPD under anti-PD-1/PD-L1 therapy, we performed the first systematic review and meta-analysis on this topic.

## 2. Results

A total of 415 potentially eligible articles was identified by the initial search (Figure 1). After the screening process, nine eligible articles were included in our analysis (Table 1, Table 2 and Appendix A) [9,10,11,12,14,15,16,17,18]. Three potentially eligible studies [13,18,19] were performed in the same institution and addressed an identical type of tumor (AGC) with a potential overlap of patients, and thus the study reporting the largest population [18] was included to avoid duplication. Nineteen patients reported by Champiat and colleagues [9] were potentially repeatedly mentioned in the two other studies [10,12], with the same type of tumors. A study by Kanjanapan and colleagues [15] included 20 patients treated with immunotherapy other than anti-PD-1/PD-L1. Nevertheless, we included these two studies [9,15] in our main analysis because the portions of these patients not of our interest in the cohorts were rather small (19 out of 119 patients, and 20 out of 170 patients, respectively), and we speculated that the data of these patients will not significantly alter the results of the analyses. To check that the data of these patients do not significantly alter the results of the main analyses, we also performed sensitivity subgroup analyses excluding the two studies.

All identified eligible studies were retrospective cohort studies. The studies were graded as fair quality (Appendix A). Three studies identified cohorts of patients with various cancers, while the other six studies identified cohorts of patients with a specific type of cancer, four of which were NSCLC. Definitions of HPD varied across the studies, and all but one study [14] assessed the acceleration of tumor growth based on three imaging timepoints to define HPD. The incidence of HPD across nine types of cancer was available for 17 studies and abstracts in addition to the eligible studies (Appendix A) and ranged from 2-30% in cohorts of patients with NSCLC, 21% for AGC, 4–29% for HNSCC, 10% for breast cancer, 8% for hepatocellular carcinoma, 1–28% for high-grade glioma, 6–10% for melanoma, 1-7% for renal cell carcinoma, and 12% for urothelial cell carcinoma.

The associations of HPD with a total of 23 potential predictive factors were suitable for meta-analysis (Table 3, Figure 2). Five factors showed statistically significant associations with HPD in patients during PD-1/PD-L1 therapy. Factors positively associated with the odds of HPD were serum lactate dehydrogenase (LDH) above the upper normal limit (odds ratio [OR] = 1.89, 95% confidence interval [CI] = 1.02–3.49, *p* = 0.043), more than two metastatic sites (OR = 1.86, 1.34–2.57, *p* < 0.001), liver metastases (OR = 3.33, 2.07–5.34, *p* < 0.001), and Royal Marsden Hospital (RMH) prognostic score [20] of 2 or above (OR = 3.33, 1.96–5.66, *p* < 0.001), whereas positive PD-L1 expression status was inversely associated with HPD (OR = 0.60, 0.36–0.99, *p* = 0.044) (Figure 2). Heterogeneity was low among the five significant associations with I^2^ ranging from 0 to 19%. 95% prediction intervals excluded the null of two characteristics (more than two metastatic sites and liver metastases), suggesting that these associations are likely to persist in future clinical settings. Small-study effect was assessed and was found in the association of PD-L1 expression, where Egger *p* value was below 0.1, and small studies tended to have greater effect sizes compared to larger studies, as shown in its funnel plot (Appendix A).

Other studied baseline characteristics were not associated with HPD, including those that were not meta-analyzed due to insufficient reporting of necessary data. Those were serum albumin [9,16,17,18], fibrinogen [9,16], C-reactive protein (CRP) [9], the number of white blood cells [16], neutrophils [9,12,16], lymphocytes [9,16,18], tumor size [9,10,15,16,17], and tumor mutational burden [9,17,18]. Exceptions were reported in a cohort of patients with AGC [18], which found higher serum CRP, higher absolute neutrophil count, and a higher sum of diameters of target lesions to be associated with increased risks of HPD [18]. These may reflect unique properties of gastric cancer or small sample size of the study. Stage of NSCLC (III vs. IV) was not associated with HPD [12,16]. Metastatic spread to sites other than the liver was not associated with HPD [16,18].

We performed subset analyses including only patients with NSCLC [12,14,16,17], and found that the results were similar with no new statistically significant association (Appendix A). Concerning liver metastases and more than two metastatic sites, 95% prediction intervals included the null in the subset analyses, probably due to the decreased number of studies. When we performed subset analyses after excluding a study by Champiat and colleagues [9] to avoid potential overlapping population and a study by Kanjanapan and colleagues [15] to exclude patients receiving treatment other than PD-1/PD-L1 inhibitors, the results were similar, with no new significant association identified (Appendix A). In both these subset analyses, the association of HPD with PD-L1 expression lost its statistical significance.

Meta-analyses of survival outcomes were not feasible due to insufficient data and heterogeneous comparison groups (Table 2). Examining the individual studies, HPD was associated with worse overall survival in NSCLC and in AGC when compared to the non-HPD group (Table 2) [14,18], but the overall survival difference was not found in patients with HNSCC [10]. In patients with NSCLC, HPD was also associated with worse overall survival when compared with the patient group with progressive disease without HPD (Table 2) [12,16,17]. Such survival differences were not found in patients who fit the criteria for HPD after receiving chemotherapy [12,13].

## 3. Discussion

We comprehensively analyzed data of 1519 patients including 217 with HPD and identified five baseline predictors associated with risks of HPD. The presence of liver metastases or more than two metastatic sites were strongly associated with higher odds of HPD with *p* values below 0.001 and 95% prediction intervals excluding the null, supporting the high accuracy of the findings. The association of serum LDH above the upper normal limit with HPD had borderline significance supported by only two studies. Nevertheless, this finding was consistent with the other three eligible studies [9,16,18] that reported higher serum LDH as a continuous variable to be associated with higher risks of HPD (median 248 U/L [HPD group] vs. 198 U/L [non-HPD group], *p* = 0.097 [9]; median 396 U/L [HPD group] vs. 180 U/L [non-HPD group], *p* = 0.006 [18,19]; median 510 U/L [HPD group] vs. 235 U/L [non-HPD group], *p* = 0.001 [16]). Higher RMH prognostic score was also associated with an increased risk of HPD. However, this association might be attributable to only two of the criteria’s components: serum LDH above the upper normal limit and more than two metastatic sites, as serum albumin < 3.5 g/dL was not associated with HPD in any of the eligible studies [9,16,17,18]. Positive PD-L1 expression was associated with lower chances of developing HPD, but at a borderline significance threshold with small-study effect. Moreover, the cut-off used for assessing PD-L1 positivity was reported in only two [12,18] out of the five studies, further contributing to the uncertainty of the association. Other clinic-pathologic factors were not found to be associated in our evidence synthesis, including factors previously reported to be associated with HPD such as advanced age [9] and oncogenic *EGFR* mutation [11]. Among the identified significant associations, effect sizes and directions were consistent across the studies with low between-study heterogeneity. In the eligible studies, HPD was observed regardless of whether the patients were given PD-1 or PD-L1 inhibitors, or whether PD-1/PD-L1 inhibitors were given as monotherapy or in combination with other therapies. Notably, HPD was also not associated with the number or type of previously received therapy, suggesting that HPD is not a phenomenon resulting from the discontinuation of the prior therapy. Nevertheless, subset analysis by these factors was not possible due to lack of data. 

Development of HPD was in general associated with poor survival. In NSCLC patients, analyses of patients with HPD exhibited lower survival rates than patients with progressive disease under immune checkpoint inhibitor therapy [12,16]. Such survival differences were not confirmed in patients who fit the criteria for HPD after receiving chemotherapy [12,13]. Hence, reduction of survival caused by HPD after anti-PD-1/PD-L1 therapy may be causative for the crossed survival curves between the immunotherapy and chemotherapy arms in several randomized trials of PD-1 [3,4,21,22] or PD-L1 inhibitors [23], where survival initially favored the chemotherapy arm in the first months, whereafter curves crossed to favor the immunotherapy arm, meaning that immunotherapy did worse than chemotherapy in the first months of these study cohorts [24]. Intriguingly, such phenomena were not found in trials comparing chemotherapy plus PD-1/PD-L1 inhibitors versus chemotherapy alone [25,26,27]. There, the rapid tumor shrinkage under chemotherapy was suggested to be present in both arms and hence to prevent the early mortality that has been associated by others with delayed responses to immune checkpoint inhibition. Nevertheless, the initial adverse effect on survival that can in some studies be observed during immune checkpoint inhibitor therapy and its potential association with HPD needs to be further investigated by future studies. The shorter survival of patients with HPD might additionally be associated with a lower number of patients receiving further treatment lines [28].

Molecular mechanisms of hyperprogression are yet to be elucidated. Modulation of the immune system by PD-1/PD-L1 inhibitors may play a role in developing HPD. Indeed, PD-1/PD-L1 blockade may potentially exert detrimental effects on the immune system via several mechanisms, including upregulation of regulatory T cells (Tregs), modulation of tumor-promoting cells, aberrant inflammation leading to angiogenesis and tissue remodeling, and activation of oncogenic pathways [24]. However, elucidation of such mechanisms in patients with HPD has only recently begun. Kamada and colleagues [19] reported that patients with HPD showed marked increases of tumor-infiltrating, highly suppressive Tregs in tumor tissue upon initiation of PD-1 inhibitor therapy, compared to their decrease in patients without HPD [19]. Lo Russo and colleagues found that HPD was associated with tumor-associated macrophages with M2-differentiation and myeloperoxidase (MPO)+ myeloid cells, and that PD-1 inhibition induced FcR triggering of M2-macrophages, promoting aggressive protumorigenic behavior [14].

A lack of pre-existing antitumor immunity may correlate with HPD after PD-1/PD-L1 inhibitor therapy. Kim and colleagues [17] reported that patients with HPD had a lower frequency of effector/memory subtype CD8+ T cells and higher frequencies of severely exhausted tumor-reactive CD8+ T cells at baseline compared to patients without HPD. Other studies also reported mutations and gene variations either directly in immune system-related genes, such as PD-1, PD-L1, *IDO,* and *VEGF*, or in tumor suppressor genes such as *TSC2* and *VHL*, along with transcriptional upregulation of oncogenic pathways in patients with HPD [14,29,30]. PD-1/PD-L1 blockade in tumors with pre-existing genetic alterations may also trigger alternative cell-intrinsic signaling pathways that promote tumorigenesis [24]. Studies also reported that oncogenic *MDM* family gene amplification and *EGFR* mutation could be associated with HPD [11], although these results were limited by their small sample sizes and not replicated in other studies [14,19] or within our meta-analysis. 

Out of the identified associations, the presence of baseline liver metastases was most strongly associated with HPD, with OR over 3 and a *p* value below 10^-6^. Liver metastases are a known negative predictor of response and prognosis of PD-1/PD-L1 inhibitor therapy, yet the exact mechanisms underlying these associations remain unclear [31,32]. Tumeh and colleagues [32] reported that patients with melanoma and liver metastases had a reduced CD8+ T cell density at the invasive tumor margin compared with patients without liver metastases, a signature associated with response to PD-1 inhibitor therapy. Lee and colleagues [33] reported a systemic influence of liver metastases on the host immune system. Besides lower CD8/Treg ratios and T-cell activation markers, they showed a decrease in tumor antigen tetramer-positive CD8 cells in mice bearing liver metastases [33]. These results suggest that liver-induced peripheral and systemic immune tolerance, which lead to better outcomes in patients receiving liver transplantation, may also induce poor responses to PD-1/PD-L1 inhibitor therapy and increased chances of HPD in patients with baseline liver metastases [32,33], although further studies are needed to confirm this hypothesis. Furthermore, patients with liver metastases may also have other baseline characteristics associated with HPD, such as multiple metastatic sites [34] or other unknown factors, resulting in a higher risk of HPD. In our analysis, the presence of more than two metastatic sites was also strongly associated with a higher risk of HPD with a *p* value below 0.001. Intriguingly, in our eligible studies, other markers of tumor burden such as baseline tumor size or stage of NSCLC were not found to be associated with HPD.

High serum LDH at baseline was associated with higher risks of HPD during PD-1/PD-L1 inhibitor therapy. Elevated serum LDH is an established poor prognostic marker of cancer [20,35]. Elevated serum LDH is known to reflect intratumor hypoxia, increased intratumor LDH activity, increased lactate production, and acidification of the extracellular space, possibly resulting in tumor cell proliferation, resistance to apoptosis, and angiogenesis [36]. An acidic tumor microenvironment is known as a potential co-factor to favor the impotence of tumor-infiltrating cytotoxic T-lymphocytes and NK-cells [37,38], and lactate stimulation in lung cancer cells induces activation of PD-L1 in tumor cells, protecting tumor cells from cytotoxic T-lymphocyte targeting [39]. These mechanisms may explain the reduced efficacy of PD-1/PD-L1 inhibitors in patients with high baseline LDH in some studies. Additionally, in hematology, LDH is associated with a high growth dynamic, hence higher tumor load and aggressiveness of the disease. However, how elevated serum LDH relates to the acceleration of tumor growth upon PD-1/PD-L1 inhibitor treatment is an unexplored issue. 

Interaction of immune checkpoint inhibitor therapies with other types of therapies in HPD remains yet unaddressed. In our meta-analysis of three studies, HPD was observed in PD-1/PD-L1 inhibitor monotherapy and in combination with other therapies. However, the small number of patients and large variability in combination therapy regimens limit the generalizability of the results. Early survival benefit seen in chemotherapy plus immunotherapy cohorts [26,27] may be due to the prevention of rapid tumor growth by chemotherapeutics, which would otherwise have led to HPD. Although this is yet a hypothesis, selectively administrating chemotherapy plus immunotherapy after identifying patients with high risk of HPD may prove to be a valid treatment strategy. Other checkpoint inhibitors such as CTLA-4 inhibitors are also known to be associated with HPD and a study reported similar HPD incidence in the PD-l/PD-L1 inhibitor and CTLA-4 inhibitor combination group and the PD-1/PD-L1 inhibitor monotherapy group [14]. Although we could not identify studies of HPD during immune checkpoint inhibitors plus radiotherapy or targeted therapies, HPD may also be observed in these combination therapies.

HPD imposes challenges in making treatment decisions for patients undergoing anti-PD-1/PD-L1 therapy. The main differential diagnostic entity of HPD is pseudoprogressive tumor, with both entities characterized by an increase in tumor burden in early imaging after the initiation of anti-PD-1/PD-L1 therapy. Biopsy of the enlarged lesion reveals abundant inflammatory cells in pseudoprogressive tumor, whereas in HPD primarily tumor cells are found [40,41]. Patients having pseudoprogressive tumor usually have a paucity of symptoms and normal performance compared to those with true progression [40]. However, the two very different response patterns currently have overlapping definitions, and it is possible to be diagnosed with both simultaneously [12], while further treatment may be beneficial in one and deleterious in the other. Early and accurate recognition of HPD is essential to enable switching to another treatment in patients under still relatively good clinical conditions. Parameters identified after treatment initiation and biopsy may also assist in differentiating between HPD and pseudoprogression. Shifts of serum laboratory data from baseline [18] and tumoral radiomic textural features [16,42] have been proposed as differentiating markers after the initiation of anti-PD-1/PD-L1 therapy.

HPD, as a recently acknowledged entity, currently lacks consensus regarding its definition (Table 2 and Appendix A). To define HPD, Champiat [9], Saada-Bouzid [10] and colleagues calculated the ratio of the rate of tumor growth (tumor growth rate [TGR] and tumor growth kinetics [TGK], respectively) based on three imaging timepoints (pre-treatment, baseline, post/under-treatment), and Ferrara and colleagues [12] utilized the absolute difference between the TGR at pre-treatment and post/under-treatment. By definition, HPD diagnosis using TGR or TGK requires the presence of measurable target lesions and at least three sequential imaging (i.e., pre-treatment, baseline, and post/under-treatment), which may not be fully available. To overcome these caveats, clinical criteria such as time-to-treatment failure [11,14] and ECOG (Eastern Cooperative Oncology Group) performance status deterioration [14] or alternative radiologic criteria such as post-therapy increase of tumor burden [14,43] have been suggested. These criteria may be useful in clinical settings to diagnose HPD in patients that did not have prior imaging and patients who receive immune checkpoint inhibitor therapy as first-line therapy. However, these clinical criteria may not be fully accurate as they use fundamentally different definitions, and studies [17,43] have reported low concordance between the two types of definitions. As HPD is distinguished from normal progression by its acceleration of tumor growth upon the initiation of therapy, comparing the speed of progression before and after therapy is necessary to precisely identify HPD. Future studies aiming to elucidate characteristics of HPD (e.g., risk factors of HPD) should define HPD using TGR or TGK based on three sequential imaging timepoints. Using either TGR ratio or TGK ratio may be acceptable, as there is a high concordance between the two definitions [17].

The variance of the definition of HPD across the eligible studies (Table 2) is also a limitation of our meta-analyses. All studies but one by Lo Russo and colleagues [14] utilized acceleration of tumor growth based on three imaging timepoints as a common parameter for defining HPD. However, additional criteria used by some studies may have resulted in heterogeneity of the identified HPD patients. This particularly relates to a criterion used by three of the eligible studies: time-to-treatment failure <2 months, which showed low concordance with criteria using TGR or TGK [17]. Additionally, the study by Lo Russo and colleagues [14] did not use pre-treatment imaging data but utilized several clinical parameters, further potentially contributing to the heterogeneity between the studies. Nevertheless, the effect direction of the individual studies in statistically significant findings was consistent and the observed heterogeneity (I^2^) was low, supporting significant associations of the predictive factors.

Other limitations also exist. The studies included in this meta-analysis had small sample sizes with few HPD cases, thus having low power to confirm associations. All identified studies had a retrospective design, and the clinic-pathologic correlations were not systematically controlled for potential confounders. Furthermore, some tests for predictive biomarkers were exploratory and not well-standardized, allowing for high chances of confounding, selection, and reporting bias. Meta-analyses were performed in heterogeneous clinical settings, particularly in terms of tumor types and treatment regimens (PD-1 inhibitors vs. PD-L1 inhibitors, monotherapy vs. combination therapy). For the listed reasons, predictability and the clinical applicability of the predictive markers identified in this meta-analysis should be further confirmed in adequately-powered, well-conducted prospective studies, and future clinical trials aiming at exploring HPD should not hastily exclude non-significant markers in our study from potentially predictive biomarkers of HPD. Associations of HPD with other factors such as microsatellite instability, tumor mutational burden, and copy number instability should additionally be explored in future studies, since these can greatly influence the response to immunotherapy [44,45].

## 4. Materials and Methods 

### 4.1. Literature Search Strategy and Eligibility Criteria

This review has been performed and reported following PRISMA guidelines (Appendix A). Two investigators (J.Y.K. and J.I.S.) independently searched PubMed, Embase, Scopus, ClinicalTrials.gov, Cochrane Controlled Register of Trials, and Web of Science from database inception to 6 June 2019 without language restriction, using keywords such as hyperprogression and PD-1/PD-L1 (full search strategy of PubMed in Appendix A). For incidence data, meeting abstracts presented at American Society of Clinical Oncology and European Society of Medical Oncology were searched. To identify eligible articles, we examined the title and the abstract, and then the full-text. We also reviewed the references of the relevant articles for potentially eligible studies. Disagreements were resolved through discussion between J.Y.K. and J.I.S.

We included prospective or retrospective studies reporting characteristics of patients who developed HPD during anti-PD-1/PD-L1 therapy, regardless of the tumor type. To be included in this review, studies had to report characteristics of cancer patients before the initiation of anti-PD-1/PD-L1 therapy separately for patient groups with HPD (HPD groups) and patient groups without HPD, which includes patients with progressive disease without HPD, stable disease, partial response, or complete response (non-HPD groups). The definition of HPD followed that of the reporting study. We excluded in vitro or animal studies, case reports, case series not reporting on non-HPD groups, and reviews. We did not include in our meta-analysis the meeting abstracts that reported data of statistically significant findings only, because including such studies may lead to selection bias. We assessed the quality of the eligible studies using the Newcastle–Ottawa Scale [46]. In the case of studies on the same patients (at least in part), we selected the manuscript presenting the largest cohort to avoid a duplicative analysis.

### 4.2. Data Extraction

We performed aggregate data meta-analyses of eligible articles. From each article, we extracted the name of the first author, publication year, the country and centers where the study was performed, type of tumor, treatment regimen (PD-1 inhibitor or PD-L1 inhibitor, given as monotherapy or combined with other therapies), study definition of HPD, the total number of patients and the number of HPD and non-HPD groups, and characteristics of the two patient groups at baseline (i.e., a timepoint prior to the initiation of PD-1/PD-L1 inhibitor therapy, but often after or alongside other therapy lines such as chemotherapy and other immunotherapies), and predictors of HPD identified in the study. We also recorded the incidence of HPD and the survival outcomes (e.g., overall survival and progression-free survival) of the HPD groups and non-HPD groups.

### 4.3. Statistical Analysis

From each included study, we first calculated the OR and its 95% CI for associations between the baseline patient characteristics and risks of HPD. Then, we meta-analyzed these individual estimates to calculate the summary OR and its 95% CI. We used a DerSimonian–Laird random-effects model [47] to account for potential heterogeneity between the studies. Heterogeneity between studies was measured by I^2^ [48]. We also estimated the 95% prediction interval, which reflects the variation in effect size over different settings [49]. We assessed small-study effects (also known as publication bias) through visual inspection of the funnel plot and Egger’s test [50]. We performed subset analyses by tumor type. All statistical tests were two-sided. We considered *p* values of 0.05 or less as significant but also assessed a lower threshold of 0.001 [51]. The software used for analyses was Revman ver.5.3.5, R ver.3.5.1, and its packages.

## 5. Conclusions

We summarized the currently available evidence on this topic, identifying potential predictive baseline factors of tumor hyperprogression. HPD during anti-PD-1/PD-L1 therapy seemed to be strongly associated with high serum LDH and the presence of more than two metastatic sites or liver metastases, but not with other baseline characteristics. Positive PD-L1 expression was associated with lower risks of developing HPD, but at borderline significance with small-study effect. Along with our results, this study highlights an unmet need for a consensus of the definition of HPD to further gain insights into this deleterious complication of immune checkpoint inhibitor therapies.

## Figures and Tables

**Figure 1 cancers-11-01699-f001:**
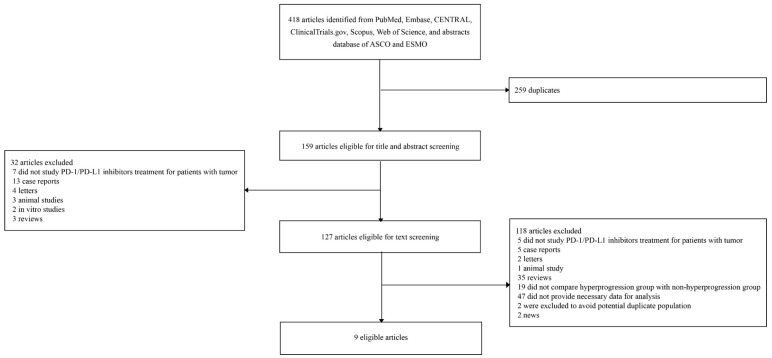
Flow chart of literature searches.

**Figure 2 cancers-11-01699-f002:**
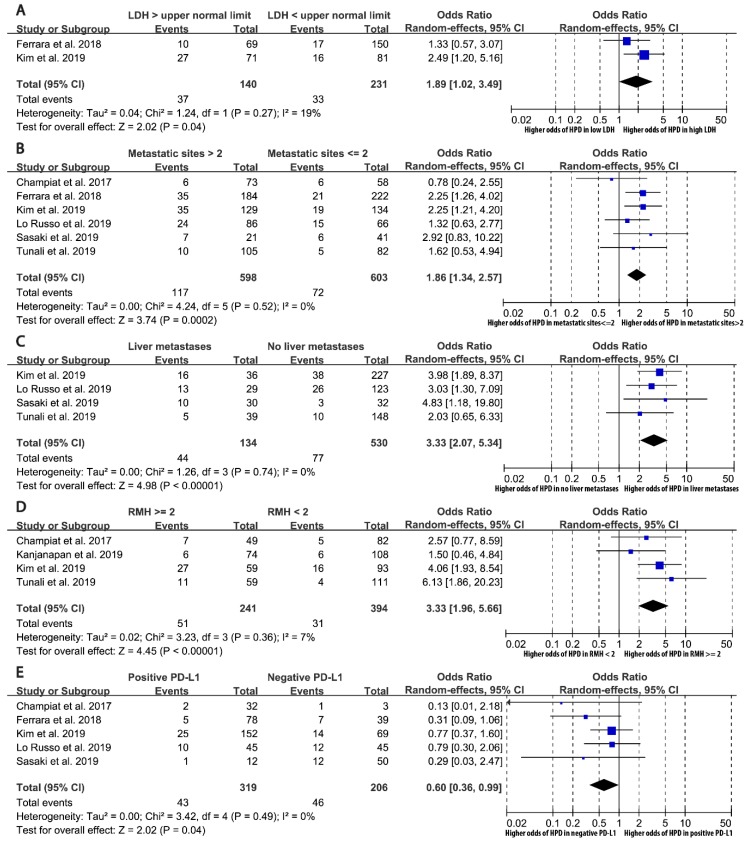
Random-effect meta-analysis forest plots representing the associations of baseline patient characteristics with hyperprogressive disease events. Only statistically significant associations are shown as forest plots. **A**. Serum lactate dehydrogenase (LDH) above the upper normal limit; **B**. More than two metastatic sites; **C**. Liver metastases; **D**. Royal Marsden Hospital (RMH) prognostic score at or above 2; **E**. Positive tumor PD-L1 expression status.

**Table 1 cancers-11-01699-t001:** Characteristics of the included studies: population characteristics.

Study	Type of Study	Country, Institution	Number of Patients	Underlying Malignancy	Treatment	Incidence of HPD
Champiat, et al. 2017 [9]	Retrospective analysis of clinical trials	France, single center	131	Melanoma (34%), lung (10%), renal (7%), colorectal (6%), urothelial (6%), others (37%)	PD-1/PD-L1 inhibitor monotherapy	9% (12/131)
Kato, et al. 2017 [11]	Retrospective cohort	USA, single center	102	NSCLC (37%), head and neck (9%), cutaneous squamous cell carcinoma (9%), melanoma (6%), renal cell carcinoma (5%)	PD-1/PD-L1 inhibitor monotherapy	6% (6/102)
Saada-Bouzid, et al. 2017 [10]	Retrospective cohort	France, 4 centers	34	Recurrent and/or metastatic head and neck squamous cell carcinoma	PD-1/PD-L1 inhibitors	29% (10/34)
Ferrara, et al. 2018 [12]	Retrospective cohort	France, 8 centers	406	NSCLC	PD-1/PD-L1 inhibitors	14% (56/406)
Lo Russo, et al. 2019 [14]	Retrospective cohort	Italy, single center	152	NSCLC	PD-1/PD-L1 inhibitors	26% (39/152)
Sasaki, et al. 2019 [18]	Retrospective cohort	Japan, single center	62	Advanced gastric cancer	Nivolumab	21% (13/62)
Kanjanapan, et al. 2019 [15]	Retrospective analysis of clinical trials	Canada, single center	182	Head and neck (18%), gynecological (16%), lung (15%), gastrointestinal (15%), genitourinary (12%), others (24%)	PD-1/PD-L1 inhibitors (89%), other checkpoint inhibitors (3%), or costimulatory molecules (8%)	7% (12/182)
Tunali, et al. 2019 [16]	Retrospective analysis of clinical trials	USA, single center	187	NSCLC	PD-1/PD-L1 inhibitors	8% (15/187)
Kim, et al. 2019 [17]	Retrospective cohort	Korea, single center	263	NSCLC	PD-1/PD-L1 inhibitors	21% (54/263)

Abbreviations: HPD, hyperprogressive disease; NSCLC, non-small cell lung cancer; PD-1, programmed cell death protein 1; PD-L1, programmed death 1 ligand 1.

**Table 2 cancers-11-01699-t002:** Characteristics of the included studies: definition, predictive factors, and prognosis of hyperprogressive disease.

Study	Definition of HPD	Predictive Factors of HPD	Impact of HPD on Overall Survival	Impact of HPD on Progression-Free Survival
Champiat, et al. 2017 [9]	• RECIST-defined PD at first evaluation and TGRpost/TGRpre^a^ ≥ 2	Advanced age of ≥ 65 years (*p* = 0.018)	HPD vs. complete response or partial response (HR 26, 95% CI 5.6–121, *p* < 0.0001)	NA
Kato, et al. 2017 [11]	• TGRpost/TGRpre^a^ ≥ 2 and > 50% increase in tumor burden and TTF < 2 months	*EGFR* mutation (*p* = 0.005), *MDM2* mutation (*p* = 0.007)	NA	NA
Saada-Bouzid, et al. 2017 [10]	• TGKpost/TGKpre^b^ ≥ 2	Regional recurrence (*p* = 0.008)	HPD vs. non-HPD (6.1 vs. 8.1 months, *p* = 0.77)	HPD vs. non-HPD (2.5 vs. 3.4 months, *p* = 0.003)
Ferrara, et al. 2018 [12]	• RECIST-defined PD at first evaluation and TGRpost−TGRpre ^a^ > 50%	Number of metastatic sites > 2 (*p* = 0.006)	HPD vs. PD without HPD (HR 2.18, 95% CI 1.29–3.69, *p* = 0.03)	NA
Lo Russo, et al. 2019 [14]	• Fulfilling at least 3 of the following 5 criteria: 1) TTF < 2 months, 2) > 50% increase in the sum of target lesions major diameters between baseline and first radiologic evaluation, 3) appearance of at least two new lesions in an organ already involved between baseline and first radiologic evaluation, 4) spread of the disease to a new organ between baseline and first radiologic evaluation, 5) ECOG ≥ 2 during the first 2 months of treatment	NA	HPD vs. non-HPD (4.4 vs. 17.7 months)	NA
Sasaki, et al. 2019 [18]	• TGKpost/TGKpre ^b^ ≥ 2 and > 50% increase in tumor burden	ECOG performance status ≥ 1 (*p* = 0.003), liver metastases (*p* = 0.029), sum of the largest diameters of target lesions ≥ median (*p* = 0.003), absolute neutrophil count ≥ median (*p* = 0.002), neutrophil-to-lymphocyte ratio ≥ median (*p* = 0.008), C-reactive protein ≥ median (*p* = 0.006), serum LDH ≥ median (*p* = 0.006)	HPD vs. non-HPD (HR 9.16, 95% CI 3.72–22.6, *p* < 0.001)	HPD vs. non-HPD (HR 4.82, 95% CI 2.36–9.57, *p* < 0.001)
Kanjanapan, et al. 2019 [15]	• RECIST-defined PD at first evaluation and TGRpost/TGRpre ^a^ ≥ 2	Female sex (*p* = 0.01)	HPD vs. non-HPD (HR 1.7, 95% CI 0.9–3.3, *p* = 0.11)	HPD vs. non-HPD (HR, 3.7, 95% CI 2.0–7.1, *p* < 0.001)
Tunali, et al. 2019 [16]	• RECIST-defined PD at first evaluation and TGRpost/TGRpre ^a^ ≥ 2 and TTF < 2 months	RMH prognostic score ≥ 2 (*p* = 0.003), higher serum LDH (*p* = 0.001)	HPD vs. PD without HPD (3.2 vs. 8.4 months, *p* < 0.001)	NA
Kim, et al. 2019 [17]	• RECIST-defined PD at first evaluation and TGRpost/TGRpre ^a^ ≥ 2 and TGKpost/TGKpre ^b^ ≥ 2	Number of metastatic sites > 2 (*p* = 0.009), liver metastases (*p* < 0.001), serum LDH > upper normal limit (*p* = 0.013), RMH prognostic score ≥ 2 (*p* = 0.002)	HPD vs. PD without HPD (HR 5.71, 95% CI 3.14–8.23, *p* < 0.05)	HPD vs. PD without HPD (HR 4.62, 95% CI 2.87–7.44, *p* < 0.05)

a. Tumor growth rate (TGR) was calculated by defining tumor size as the sum of the longest diameters of the target lesions as per the RECIST criteria, and by assuming the tumor growth follows an exponential law, as described extensively in Champiat, et al. 2017 [9]. TGRpost/TGRpre stands for the ratio of TGR after the initiation of experimental treatment to TGR before the initiation of experimental treatment. TGRpost−TGRpre > 50% stands for an absolute increase in the TGR exceeding 50% per month. b. Tumor growth kinetics (TGK) was calculated by defining tumor size as the sum of the longest diameters of the target lesions as per the RECIST criteria, and by assuming a linear tumor growth model, as described extensively in Saada-Bouzid, et al. 2017 [10]. TGKpost/TGKpre stands for the ratio of TGK after the initiation of experimental treatment to TGK before the initiation of experimental treatment. Abbreviations: CI, confidence interval; ECOG, Eastern Cooperative Oncology Group; HPD, hyperprogressive disease; HR, hazard ratio; LDH, lactate dehydrogenase; NA, not available; PD, progressive disease; RMH, Royal Marsden Hospital; TGK, tumor growth kinetics; TGR, tumor growth rate; TTF, time-to-treatment failure.

**Table 3 cancers-11-01699-t003:** Results of random-effects meta-analyses of associations between baseline patient characteristics and odds of HPD.

Baseline Patient Characteristics		Number of Study Estimates	Number of HPD/Non-HPD Patients	Random-Effects Summary Estimate, Odds Ratio and 95% Confidence Interval ^a^	*p* Value ^a^	I^2^	95% Prediction Interval ^b^	Egger *p* Value ^b^
Age ≥ 65		5	102/837	0.86 (0.56–1.32)	0.49	0%	0.43–1.73	0.25
Female sex		7	155/856	1.34 (0.74–2.46)	0.34	49%	0.27–6.63	0.68
Smoking history		5	174/859	0.78 (0.50–1.22)	0.28	0%	0.38–1.61	0.18
ECOG performance status ≥ 2		3	40/391	2.76 (0.83–9.13)	0.096	52%	<0.01–690,014.13	0.70
ECOG performance status ≥ 1		3	149/672	1.14 (0.63–2.04)	0.67	24%	0.01–175.13	0.0075
Neutrophil-to-lymphocyte ratio ≤ 3		2	85/463	0.89 (0.55–1.43)	0.62	0%	NA	NA
Serum lactate dehydrogenase > upper normal limit		2	70/301	**1.89 (1.02–3.49)**	**0.043**	19%	NA	NA
Number of metastatic sites > 2		6	189/1012	**1.86 (1.34–2.57)**	**0.00018**	0%	1.17–2.94	0.41
Liver metastases		4	121/543	**3.33 (2.07–5.34)**	**0.00000064**	0%	1.18–9.40	0.86
RMH prognostic score ≥ 2		4	82/553	**3.33 (1.96–5.66)**	**0.0000086**	7%	0.88–12.58	0.64
PD-L1 positive		5	89/436	**0.60 (0.36–0.99)**	**0.044**	0%	0.26–1.35	0.056
PD-1 inhibitor vs. PD-L1 inhibitor		6	182/913	1.22 (0.76–1.94)	0.41	0%	0.63–2.36	0.33
Combination therapy vs. monotherapy		3	107/632	1.98 (0.46–8.57)	0.36	67%	<0.01-27,517,080.41	0.77
Previous treatment lines > 2		4	162/721	1.17 (0.81–1.68)	0.40	7%	0.47–2.91	0.49
Previous chemotherapy		4	137/850	1.14 (0.72–1.81)	0.58	3%	0.38–3.40	0.18
Previous radiotherapy		5	106/714	0.77 (0.33–1.83)	0.56	0%	0.19–3.14	0.44
Previous targeted therapy		4	137/850	1.40 (0.84–2.32)	0.20	1%	0.45–4.34	0.90
Previous immunotherapy		2	68/469	2.25 (0.67–7.56)	0.19	0%	NA	NA
Previous corticosteroid		2	27/291	1.92 (0.67–5.49)	0.23	0%	NA	NA
NSCLC	*EGFR* mutation	5	136/759	1.29 (0.48–3.52)	0.61	37%	0.09–19.35	0.57
	*KRAS* mutation	2	46/197	0.59 (0.19–1.83)	0.36	0%	NA	NA
	*ALK* rearrangement	3	121/538	2.86 (0.65–12.52)	0.16	0%	<0.01–41,535.21	0.15
	Squamous histology	4	164/844	0.87 (0.58–1.31)	0.50	10%	0.30–2.53	0.82

a. Statistically significant associations are shown in bold. All statistical tests are two-sided. b. Not available for meta-analyses of two studies. Abbreviations: ECOG, Eastern Cooperative Oncology Group; HPD, hyperprogressive disease; NA, not available; NSCLC, non-small cell lung cancer; PD-1, programmed cell death protein 1; PD-L1, programmed death 1 ligand 1; RMH, Royal Marsden Hospital.

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
