# Peer review of "Hyperprogressive Disease during Anti-PD-1 (PDCD1) / PD-L1 (CD274) Therapy: A Systematic Review and Meta-Analysis"

_cancers, 2019, doi:10.3390/cancers11111699_

Round 1
Reviewer 1 Report
Jong Yeob Kim et al. conduct the study by using systemic review and meta-analysis to identify the clinical factors which associate with Hyperprogressive disease (HPD) in cancer patients receiving anti-PD-1/PD-L1 inhibitors. In general, this mauscript was organized and written well. The main five factors associated with HPD in anti-PD-1/PD-L1 therapy identified by this study are LDH, more than two metastatic sites, liver metastases, Royal Marsden Hospital prognostic 60 score of 2 or above, and positive PD-L1 expression status. The findings of this study may have some impacts on clinical practice while using anti-PD-1/PD-L1 immune checkpoint inhibitors. I only have a minor concern to their discussion.
It is better to add some of their own opinions and discussion on the future perspectives about other therapies (chemotherapy, radiotherapy or molecular therapies…etc.) in combination with anti-PD-1/PD-L1 immune checkpoint inhibitors in cancer patients with risk factors of HPD.
Author Response
Jong Yeob Kim et al. conduct the study by using systemic review and meta-analysis to identify the clinical factors which associate with Hyperprogressive disease (HPD) in cancer patients receiving anti-PD-1/PD-L1 inhibitors. In general, this mauscript was organized and written well. The main five factors associated with HPD in anti-PD-1/PD-L1 therapy identified by this study are LDH, more than two metastatic sites, liver metastases, Royal Marsden Hospital prognostic 60 score of 2 or above, and positive PD-L1 expression status. The findings of this study may have some impacts on clinical practice while using anti-PD-1/PD-L1 immune checkpoint inhibitors. I only have a minor concern to their discussion.
-> Thank you very much for taking your time to review the manuscript, and for your positive comments acknowledging the organization of our manuscript and the significance of our findings in this emerging field of cancer immunotherapy.
It is better to add some of their own opinions and discussion on the future perspectives about other therapies (chemotherapy, radiotherapy or molecular therapies…etc.) in combination with anti-PD-1/PD-L1 immune checkpoint inhibitors in cancer patients with risk factors of HPD.
-> Thank you for pointing out an important issue that should be addressed in detail. Now in a paragraph in the discussion section, we speculate the possibilities of HPD in combination with other therapies. The paragraph reads:
“Interaction of immune checkpoint inhibitor therapies with other types of therapies remains yet unaddressed. In our meta-analysis of three studies, HPD was observed in PD-1/PD-L1 inhibitor monotherapy and in combination with other therapies. However, the small number of patients and large variability in combination therapy regimens limit the generalizability of the results. Early survival benefit seen in chemotherapy plus immunotherapy cohorts [26,27] may be due to the prevention of rapid tumor growth by chemotherapeutics, which would otherwise have led to HPD. Although this is yet a hypothesis, selectively administrating chemotherapy plus immunotherapy after identifying patients with high risk of HPD may prove to be a valid treatment strategy. Other checkpoint inhibitors such as CTLA-4 inhibitors are also known to be associated with HPD and a study reported similar HPD incidence in the PD-l/PD-L1 inhibitor and CTLA-4 inhibitor combination group and the PD-1/PD-L1 inhibitor monotherapy group [14]. Although we could not identify studies of HPD during immune checkpoint inhibitors plus radiotherapy or targeted therapies, HPD may also be observed in these combination therapies.”
Reviewer 2 Report
Manuscript 622430, submitted to Cancers MDPI, by Kim et. al. provides novel insights into baseline patient factors that correlate with hyperprogessive disease (HPD) after immune checkpoint inhibitor cancer therapy. The presence of liver metastases appears to clearly predict HPD, especially compared to some of the other putative predictive biomarkers. Surprisingly the type of prior cancer treatments did not appear to be associated with future occurrence of HPD.The formal characterization of HPD is a relatively new field, making this study an important and timely addition to the literature, because it has the potential to influence fundamental and translational research, as well as clinical decisions that are made regarding the use of a routinely-used immunotherapy. I therefore endorse the publication of this manuscript after the authors have addressed the points below.
Strengths
The authors used a logical and clear approach to select and then perform a meta-analysis of individual research studies. The reasoning and interpretation of the data strongly suggest a links between checkpoint inhibitor therapy and HPD in varied percentage of patients, and importantly defined 5 additional baseline parameters as predictive biomarkers of HPD These putative biomarkers could now be confirmed in larger future prospective studies. The data is represented in a simple and accessible manner, and the study comprises of a very good level of detail. The scientific method is meticulous, including comprehensive supplementary material e.g. PRISMA form, that thoroughly documents the main aspects of the approach and analysis.
Weaknesses
As the authors point out, there is currently no standard or completely unanimous definition of HPD, which has ramifications in the overall interpretation of the meta-analysis. For example, the global findings and conclusions reached from the individual studies have an intrinsic and strong connection to the definition of HPD. Hence incorporating this ‘incommensurate’ data of diverse definitions of HPD into a single meta-analysis may blur or alter the results. Furthermore, this effect is expected to reduce the ability of the authors to make precise conclusions about the impact of HPD in these cohorts. However, because a lack of perfectly matched experimental design is a common problem in meta-analyses, my opinion is that the authors made an acceptable effort to acknowledge this and interpret their findings with an appropriate level of caution.
Items that require rectification before publication
Please describe how you have addressed these items in a ‘reply to the reviewer’ document, and also highlight in the updated manuscript (either by highlighted text or the Microsoft Word track changes feature) where these modifications have been inserted.
Discuss how the varied definitions of HPD in individual studies may limit the current findings, provide an opinion or hypothesis about the type(s) of definitions that appear to be the most useful/precise one in the primary research studies as well as meta-analyses. Forest plots Spell out what M-H means at least once somewhere in the manuscript (Mantel Haenszel I’m assuming), although it would also be useful in the Table notes/figure legends wherever appropriate add more information to the x axis add more numerals and ticks on all of the x axes so that it is possible to visually deduce the approximate value of each diamond or square add x axis label on each side of the line of no effect, e.g. ‘favours HPD’ vs ‘does not favour HPD’. Line 121, please provide a short rationale statement for the inclusion of the small cohort studies for some analyses but not for the sensitivity analyses. Careful editing is required to improve the English grammar and remove small errors like incorrectly-positioned full-stops, which I noticed in several places in the manuscript.
Optional Recommendations
Additional studies from after 6thJuly 2019 could be included in order to make the meta-analysis as up-to-date as possible In the discussion section, you could mention which parameters might be the very beneficial to include in clinical trials aimed at exploring HPD, yet are often overlooked or are not standard practice at the moment. Consider changing the word ‘baseline’ to something else (e.g. ‘reference’ is used in Chamionat 2017) because a substantial number of patients in the analysis already had or were having some form of cancer therapy. Alternatively, clearly define 'baseline' somewhere as ‘a timepoint immediately prior to PD1 immunotherapy, but often after or alongside other immune interventions in combination therapy’. Line 290, ‘an acidic condition’, consider changing to ‘an acidic tumor microenvironment’ because ‘acidic condition’ on its own is ambiguousAuthor Response
Manuscript 622430, submitted to Cancers MDPI, by Kim et. al. provides novel insights into baseline patient factors that correlate with hyperprogessive disease (HPD) after immune checkpoint inhibitor cancer therapy. The presence of liver metastases appears to clearly predict HPD, especially compared to some of the other putative predictive biomarkers. Surprisingly the type of prior cancer treatments did not appear to be associated with future occurrence of HPD. The formal characterization of HPD is a relatively new field, making this study an important and timely addition to the literature, because it has the potential to influence fundamental and translational research, as well as clinical decisions that are made regarding the use of a routinely-used immunotherapy. I therefore endorse the publication of this manuscript after the authors have addressed the points below.
-> Thank you very much for taking your valuable time to review the manuscript, and for your positive comments acknowledging the significance of our findings in this emerging field of cancer immunotherapy. We also thank you for your insightful comments regarding various aspects of the study that overall helped us improve the quality of the manuscript.
Strengths
The authors used a logical and clear approach to select and then perform a meta-analysis of individual research studies. The reasoning and interpretation of the data strongly suggest a links between checkpoint inhibitor therapy and HPD in varied percentage of patients, and importantly defined 5 additional baseline parameters as predictive biomarkers of HPD These putative biomarkers could now be confirmed in larger future prospective studies. The data is represented in a simple and accessible manner, and the study comprises of a very good level of detail. The scientific method is meticulous, including comprehensive supplementary material e.g. PRISMA form, that thoroughly documents the main aspects of the approach and analysis.
-> Thank you so much again for highlighting the significance of our findings and the sound approach we took in constructing our manuscript. By further addressing your advice regarding methodological details and data presentation, we believe it helped the manuscript to be more accessible and intuitive for the readers.
Weaknesses
As the authors point out, there is currently no standard or completely unanimous definition of HPD, which has ramifications in the overall interpretation of the meta-analysis. For example, the global findings and conclusions reached from the individual studies have an intrinsic and strong connection to the definition of HPD. Hence incorporating this ‘incommensurate’ data of diverse definitions of HPD into a single meta-analysis may blur or alter the results. Furthermore, this effect is expected to reduce the ability of the authors to make precise conclusions about the impact of HPD in these cohorts. However, because a lack of perfectly matched experimental design is a common problem in meta-analyses, my opinion is that the authors made an acceptable effort to acknowledge this and interpret their findings with an appropriate level of caution.
-> Thank you for highlighting the important point that must be taken into account in interpreting our meta-analysis, and for acknowledging our efforts to advise the readers to take cautious approach in interpreting our findings. We sincerely hope that our findings will help conduction of future prospective studies that account for the potential bias not fully addressed by our manuscript to draw firmer conclusion regarding the risk factors of HPD.
Items that require rectification before publication
Discuss how the varied definitions of HPD in individual studies may limit the current findings
-> Thank you for your thoughtful advice. We now dedicated a paragraph describing in detail how the heterogeneity of the definition of HPD across the individual studies may limit the current findings. The paragraph reads: “The variance of the definition of HPD across the eligible studies (table 2) is also a limitation of our meta-analyses. All studies but one by Lo Russo and colleagues [14] utilized acceleration of tumor growth based on three imaging timepoints as a common parameter for defining HPD. However, additional criteria used by some studies may have resulted in heterogeneity of the identified HPD patients. This particularly relates to a criterion used by three of the eligible studies: time-to-treatment failure < 2 months, which showed low concordance with criteria using TGR or TGK [17,44]. Additionally, the study by Lo Russo and colleagues [14] did not use pre-treatment imaging data but utilized several clinical parameters, further potentially contributing to the heterogeneity between the studies. Nevertheless, the effect direction of the individual studies in statistically significant findings was consistent and the observed heterogeneity (I2) was low, supporting significant associations of the predictive factors.”
provide an opinion or hypothesis about the type(s) of definitions that appear to be the most useful/precise one in the primary research studies as well as meta-analyses.
-> We appreciate your advice. In the discussion section, we now describe in more detail the considerations by addressing the HPD definition into two categories: one, clinically useful criteria, and two, precise criteria that should be used in HPD research. Modified/added sentences are: “These criteria may be useful in clinical settings to diagnose HPD in patients that did not have prior imaging and patients who receive immune checkpoint inhibitor therapy as first-line therapy. However, these clinical criteria may not be fully accurate as they use fundamentally different definitions, and studies [17,43] have reported low concordance between the two types of definitions. As HPD is distinguished from normal progression by its acceleration of tumor growth upon the initiation of therapy, comparing the speed of progression before and after therapy is necessary to precisely identify HPD. Future studies aiming to elucidate characteristics of HPD (e.g. risk factors of HPD) should define HPD using TGR or TGK based on three sequential imaging timepoints. Using either TGR ratio or TGK ratio may be acceptable, as there is a high concordance between the two definitions [17].”
Forest plots Spell out what M-H means at least once somewhere in the manuscript (Mantel Haenszel I’m assuming), although it would also be useful in the Table notes/figure legends wherever appropriate add more information to the x axis add more numerals and ticks on all of the x axes so that it is possible to visually deduce the approximate value of each diamond or square add x axis label on each side of the line of no effect, e.g. ‘favours HPD’ vs ‘does not favour HPD’.
-> We thank you for your important comment regarding the design of figure 2. Fully taking your advice, we have extensively re-designed figure 2. We have removed the term “M-H” (i.e. Mantel Haenszel) because the term is not used in the manuscript and replaced it with “Random-effects” which is more consistent with the manuscript. On the x axis, we added additional numerals (2, 5, 0.5, 0.2) and added ticks that would help readers approximate the values on the figure plot. We added label on each side of the line, such as “Higher odds of HPD in liver metastases.”
Line 121, please provide a short rationale statement for the inclusion of the small cohort studies for some analyses but not for the sensitivity analyses.
-> Thank you for your comments that helped us clarify the statistical details of our study. Following your advice, we have changed the sentence to “To check that the data of these patients do not significantly alter the results of the main analyses, we also performed sensitivity subgroup analyses excluding the two studies.” and modified the former sentence to be more detailed.
Careful editing is required to improve the English grammar and remove small errors like incorrectly-positioned full-stops, which I noticed in several places in the manuscript.
-> Thank you for recognizing and pointing out grammatical and formative errors of our manuscript. We have extensively identified and corrected the grammar issues and incorrectly-positioned full-stops throughout the manuscript.
Optional Recommendations
Additional studies from after 6thJuly 2019 could be included in order to make the meta-analysis as up-to-date as possible
-> We appreciate your recommendation that is of great importance for meta-analyses. Following the recommendation, we have searched for databases such as PubMed, Embase, and CENTRAL for potentially eligible studies from after July 6th, 2019. However, we were not able to identify any additional eligible studies. One close study was a HPD retrospective cohort study (Kim, et al. PMID: 31195179), but we were not able to include this study because this study only reported baseline characteristics of patients who showed HPD and patients who showed PD but not HPD, and did not report characteristics of patients who did not show HPD nor PD.
In the discussion section, you could mention which parameters might be the very beneficial to include in clinical trials aimed at exploring HPD, yet are often overlooked or are not standard practice at the moment.
-> Thank you for your thoughtful comments. We speculate that parameters related to poor prognosis of cancer patients (especially after immunotherapy) are potential candidates of HPD because HPD is related to low survival and very poor prognosis. However, it may be hard to specify which of these markers can actually predict HPD, especially because non-significant predictors identified in our studies were consistently non-significant across the included studies. Nevertheless, to ensure that future clinical trials may not hastily exclude the non-significant predictive markers identified in our study from the candidates, we now state more clearly with subject in the discussion: “future clinical trials aiming at exploring HPD should not hastily exclude non-significant markers in our study from potentially predictive biomarkers of HPD.”
Consider changing the word ‘baseline’ to something else (e.g. ‘reference’ is used in Chamionat 2017) because a substantial number of patients in the analysis already had or were having some form of cancer therapy. Alternatively, clearly define 'baseline' somewhere as ‘a timepoint immediately prior to PD1 immunotherapy, but often after or alongside other immune interventions in combination therapy’.
-> Thank you for your insightful comment. We fully agree with your point that the term ‘baseline’ may arouse confusion to the readers, and thus made it clear in the methods section that ‘baseline’ is defined as “a timepoint prior to the initiation of PD-1/PD-L1 inhibitor therapy, but often after or alongside other therapy lines such as chemotherapy and other immunotherapies”, and also added a simpler definition in the abstract.
Line 290, ‘an acidic condition’, consider changing to ‘an acidic tumor microenvironment’ because ‘acidic condition’ on its own is ambiguous
-> Thank you for pointing out the ambiguity. We have changed the term as your suggestion.